# Isolation of Cysteine-Rich Peptides from *Citrullus colocynthis*

**DOI:** 10.3390/biom10091326

**Published:** 2020-09-16

**Authors:** Behzad Shahin-Kaleybar, Ali Niazi, Alireza Afsharifar, Ghorbanali Nematzadeh, Reza Yousefi, Bernhard Retzl, Roland Hellinger, Edin Muratspahić, Christian W. Gruber

**Affiliations:** 1Center for Physiology and Pharmacology, Medical University of Vienna, 1090 Vienna, Austria; shahin.bio65@gmail.com (B.S.-K.); bernhard.retzl@meduniwien.ac.at (B.R.); roland.hellinger@meduniwien.ac.at (R.H.); edin.muratspahic@meduniwien.ac.at (E.M.); 2Department of Plant Biotechnology, Shiraz University, Shiraz 7144165186, Iran; niazi@shirazu.ac.ir; 3Department of Plant Protection, Shiraz University, Shiraz 7144165186, Iran; afshari@shirazu.ac.ir; 4Department of Plant Breeding and Biotechnology, SANRU, Sari P.O. Box 578, Iran; gh.nematzadeh@sanru.ac.ir; 5Department of Biology, Shiraz University, Shiraz 7194684795, Iran; ryousefi@shirazu.ac.ir

**Keywords:** Cucurbitaceae, *Citrullus colocynthis*, peptides, trypsin inhibitor, knottin

## Abstract

The plant *Citrullus colocynthis*, a member of the squash (Cucurbitaceae) family, has a long history in traditional medicine. Based on the ancient knowledge about the healing properties of herbal preparations, plant-derived small molecules, e.g., salicylic acid, or quinine, have been integral to modern drug discovery. Additionally, many plant families, such as Cucurbitaceae, are known as a rich source for cysteine-rich peptides, which are gaining importance as valuable pharmaceuticals. In this study, we characterized the *C. colocynthis* peptidome using chemical modification of cysteine residues, and mass shift analysis via matrix-assisted laser desorption ionization time-of-flight (MALDI-TOF) mass spectrometry. We identified the presence of at least 23 cysteine-rich peptides in this plant, and eight novel peptides, named citcol-1 to -8, with a molecular weight between ~3650 and 4160 Da, were purified using reversed-phase high performance liquid chromatography (HPLC), and their amino acid sequences were determined by de novo assignment of b- and y-ion series of proteolytic peptide fragments. In silico analysis of citcol peptides revealed a high sequence similarity to trypsin inhibitor peptides from *Cucumis sativus*, *Momordica cochinchinensis*, *Momordica macrophylla* and *Momordica sphaeroidea*. Using genome/transcriptome mining it was possible to identify precursor sequences of this peptide family in related Cucurbitaceae species that cluster into trypsin inhibitor and antimicrobial peptides. Based on our analysis, the presence or absence of a crucial Arg/Lys residue at the putative P1 position may be used to classify these common cysteine-rich peptides by functional properties. Despite sequence homology and the common classification into the inhibitor cysteine knot family, these peptides appear to have diverse and additional bioactivities yet to be revealed.

## 1. Introduction

Small molecules have been the dominant class of chemicals in drug discovery and development, in particular due to their favorable pharmacokinetic properties and low production costs [1]. However, they often lack target selectivity, which is generally associated with unwanted side-effects in clinical applications. Hence, more recently, biologics such as antibodies have emerged as promising molecular entities in drug development since they are highly selective for their target, but they have high production costs. One drawback associated with using proteins as pharmaceutical drugs is their low metabolic and structural stability [2,3]. To overcome these limitations, small peptides (~5–50 amino acids in length) appear to be ideal molecules: peptide and medicinal chemists are able to design peptides with drug-like properties, such as metabolic stability, great efficacy and low off-target effects [4]. Interestingly, nature has provided the blueprints for many of these molecules. Nature-derived cyclic and/or cysteine-stabilized peptides are an emerging class of natural products as templates for drug development or as drug lead candidates [5,6]. In fact, there are numerous clinical candidates and approved peptide drugs available that were derived from natural sources [7,8,9]. To highlight two examples: the cone snail toxin-derived ziconotide [10] has been on the market since 2000 for management of neuropathic pain, and the plant-derived peptide T20K is in clinical development for multiple sclerosis [11].

Plants provide ’natural libraries’ for the discovery of novel cysteine-stabilized peptides (also known as cysteine-rich peptides, CRPs). They are still an underexplored class of molecules, which exhibit great natural diversity [12]. They are gene-encoded and ribosomally produced natural products [13] and hence they are accessible not only by chemical extraction and mass spectrometry, but via genetic mining and bioinformatics analysis [14]. Their endogenous physiological function in planta appears to be defense against herbivores and microbes, and therefore CRPs have demonstrated several pharmacological activities including anthelmintic [15], antimalarial [16], antimicrobial [17], antiviral [18], antitumor [19], and protease inhibition [20], which makes them valuable molecules for drug discovery. Structurally, the disulfide bonds of CRPs are often arranged in a knotted configuration, which provides a stable structural fold [21]. Hence, CRPs have received much attention as templates or scaffolds for designing orally active drug lead molecules since they exhibit high resistance to the acidic pH conditions and proteolytic environment of the gastrointestinal tract [22].

In the present study, we explored the diversity of CRPs from *Citrullus colocynthis* (Cucurbitaceae). The Cucurbitaceae family is well known as a rich source of CRPs containing an inhibitor cystine knot (ICK) structural motif [23,24,25,26]. The species *C. colocynthis* (known under the common names bitter apple, desert gourd or vine of Sodom) is a desert vine plant native to the Mediterranean Basin and Asia. It is well known for its anti-inflammatory [27] and antidiabetic properties in evidence-based traditional medicine [28,29]. In addition, some isolated constituents of this plant like cucurbitacin have been explored as therapeutic option for cancer treatment: it showed growth inhibitory activity on human breast cancer cells [30] and cytotoxicity on multidrug-resistant cancer cells [31].

Following chemical extraction, we used mass spectrometry to characterize CRPs in *C. colocynthis*. We identified at least 23 different ’citcol’ peptides. Several novel peptides were isolated for analysis of their amino acid sequence. Since they appeared to be homologous to other known squash trypsin inhibitors, we attempted to confirm their function using enzyme inhibition assays and used genome/transcriptome mining to identify structural features in Cucurbitaceae precursor sequences to classify this family of CRPs.

## 2. Materials and Methods

### 2.1. Plant Material and Extraction

The aerial parts of C. colocynthis (L.) Schrad. were collected in the summer of 2018 in the region around Toodej mountain of Estahban city in Fars Province, Iran at an altitude of 1300 m. A voucher specimen (no. 3842) was deposited and identified by the Agricultural Research Education and Extension Organization Institute, Iran. The plant material was dried and stored at 25 °C in a moisture-free environment. The dried seeds and fruits of C. colocynthis (400 g) were ground using a coffee grinder and, peptides were extracted with 2 L of dichloromethane/methanol (1:1; *v*/*v*) by continuous agitation overnight at 25 °C [32]. Next, the extract was filtered, and the peptide-rich fraction was separated by liquid–liquid fractionation by adding 1 L of ddH_2_O. The remaining aqueous extract was concentrated using a rotary evaporator and dried by lyophilization. The crude extract was dissolved in 5% of solvent B (90% acetonitrile, 9.9% ddH_2_O and 0.1% trifluoroacetic acid, TFA, *v*/*v*/*v*) and peptides were enriched by solid-phase extraction with C_18_ silica beads (ZEOprep 60 Å, C_18_ irregular 40–63 µm; ZEOCHEM, Uetikon, Switzerland), after activating with one bed volume of methanol and equilibration with two bed volumes of solvent A (99.9% ddH_2_O and 0.1% TFA, *v*/*v*). Plant compounds bound to the C_18_ material were washed and eluted with increasing concentrations of solvent B (wash 1:10%, wash 2:25%, elution 1:35% and elution 2:70%). The resulting eluates were analyzed by matrix-assisted laser desorption/ionization–time of flight (MALDI-TOF) mass spectrometry (MS) as outlined below. The peptide-rich fraction eluting between 35–70% solvent B was lyophilized and stored at −20 °C until further use.

### 2.2. Reversed-Phase High Performance Liquid Chromatography (RP-HPLC) Purification

A Dionex Ultimate 3000 HPLC system (Thermo Fisher Scientific, Waltham, MA) was used for further fractionation and purification. Aliquots of the peptide-rich fraction (100 mg) were applied and purified in multiple cycles on a preparative Kromasil C_18_ column (dichrom GmbH, Marl, Germany; 250 × 21.2 mm, 10 µm, 110 Å), at a flow rate set to 8 mL/min with up to 75-min linear gradient (0.8%/min) from 10% (5 min) to 66% (75 min) of solvent C (90% acetonitrile, 9.92% ddH_2_O and 0.08% TFA, *v*/*v*/*v*). The peptide-containing fractions were identified by MS-analysis and further purified by semi-preparative RP-HPLC (Kromasil C_18_ column, 250 × 10 mm, 5 μm) with a flow rate of 3 mL/min, and a linear gradient of 0.7%/min of solvent C. Elution profiles were monitored via a multiwavelength detector at 214, 254, and 280 nm and peaks were collected manually. The identity and purity of the peptides were determined by MALDI-TOF analysis and analytical HPLC using a Kromasil C_18_ column (5 µm, 100 Å, 250 × 4.6 mm) with a flow rate of 1 mL/min.

### 2.3. Reduction, Alkylation and Enzymatic Digestion of Peptides

The purified and lyophilized peptides (50 μg each) were dissolved in 0.2 mL of 100 mM ammonium bicarbonate/NaOH buffer, pH 8.6. The mixture was incubated for 30 min after addition of 12 μL freshly prepared dithiothreitol (DTT, Sigma-Aldrich, St. Louis, MO; final concentration: 10 mM), at 60 °C in the absence of light under gentle agitation. Iodoacetamide (Sigma-Aldrich, St. Louis, MO) was added to the reaction mixture at a final concentration of 50 mM and incubated for 1 min at 65 °C. Quenching of unreacted iodoacetamide was performed by adding 5.5 μL DTT solution (final concentration of 5 mM), then the mixture was diluted two-fold by adding ammonium bicarbonate buffer. The reduced and alkylated peptides (50 μg) were enzymatically digested by 0.5 μg of endoproteinase Glu-C, 0.2 μg of trypsin, or 0.5 μg of chymotrypsin (all proteases were MS-grade from Sigma-Aldrich, St. Louis, MO, USA) at 37 °C for 3 to 16 h, and the digestion was monitored over time by recording MALDI-TOF spectra in two hour intervals. The digestion was terminated by adding TFA at a final concentration of 0.3%, the samples were desalted and concentrated by using C_18_ Zip-Tips^TM^ according to manufacturer’s instructions (Millipore, Billerica, MA) and stored at −20 °C prior to analysis.

### 2.4. MALDI-TOF/TOF Analysis and De Novo Peptide Sequencing

Molecular weight analysis of peptides was performed by MALDI-TOF mass spectrometry on a 4800 Analyzer (ABSciex, Framingham MA, USA) operated in reflector positive ion mode acquiring between 2000 and 10,000 total shots per spectrum with a laser intensity set between 4500 and 4800. The matrix α-cyano-4-hydroxycinnamic acid (Sigma–Aldrich, St. Louis, MO, USA) was prepared by dissolving 5 mg of the matrix in 1 mL of 50% acetonitrile containing 0.1% TFA (*v*/*v*), and 0.5 μL of the samples were directly spotted onto the target plate after mixing in a 1:6 ratio (sample: matrix). Mass analysis of native plant extract and peptide-enriched fractions were acquired and processed using 4800 Analyzer Software in the range of 500–5000 *m/z* and analyzed by Data Explorer Software. Corresponding precursor masses were used for tandem mass fragmentation in positive ionization mode, with/without collision-induced dissociation with laser energy of 1 kV. Finally, the sequences were determined by manual fragment assembly of identified *N*-terminal b- and *C*-terminal y-ions series. The isobaric amino acids leucine and isoleucine as well as glutamine and lysine were distinguished by identification of site-specific cleavage fragments of trypsin and chymotrypsin digestions. The determined amino acid sequences were confirmed using the ion fragmentation calculator in Data Explorer software, sequence homology and amino acid composition analysis (described below).

### 2.5. Amino Acid Composition Analysis

For confirmation of peptide sequences High Sensitivity Amino Acid Analysis (Macquarie University, Sydney, Australia) was performed. In brief, the lyophilized peptides were dissolved in 0.2 mL of 20% acetonitrile/0.1% TFA (*v*/*v*) and aliquots were dried for analysis. Samples underwent 24 h gas phase hydrolysis in 6 M hydrochloric acid at 110 °C. After hydrolysis, all amino acids were labelled using the Waters AccQTag Ultra chemistry (following supplier’s recommendations) and analyzed on a Waters Acquity UPLC. Aliquots were analyzed in duplicate and results expressed as an average (units of ng/sample).

### 2.6. Trypsin, α-Chymotrypsin and α-Amylase Inhibitory Assays

The inhibitory assays were performed in 96-well microtiter plates using a Synergy H4 Hybrid reader (BioTek). For the trypsin inhibitor assay, the reaction mixture contained 140 µL of 20 mM Tris-HCl buffer (pH 8.0), 20 µL bovine pancreatic trypsin (Sigma Aldrich) at a final concentration of 2.5 μg/well, 20 µL citcol peptides at different concentrations and 20 µL of the substrate benzoyl-arginine-p-nitroanilide dissolved in 20 mM Tris-HCl buffer (pH 8.0) containing 20% dimethylsulfoxide. The mixture was equilibrated for 10 min at 5 °C and then the substrate was added automatically (900 µM final concentration) at the start of the assay (time point 0). The absorbance was recorded at a wavelength of 410 nm in 5 min intervals for 1 h at 37 °C. The α-chymotrypsin inhibitory assay was performed with 1 μg/well of pancreatic bovine α-chymotrypsin (Sigma Aldrich) and Ala-Ala-Phe-7-amino-4-methylcoumarin (150 µM final concentration) as substrate similar to the trypsin inhibitor assay with the difference that the fluorescence signal was recorded at 480±20 nm after excitation at 360 ± 20 nm. For the α-amylase inhibition assay 50 µL of citcol peptides at different concentrations were added to 150 µL of 20 mM phosphate buffer (pH 6.9) containing 6.7 mM NaCl and 2 U of α-amylase from *Aspergillus oryzae*, *Bacillus subtilis* or *Sus scrofa*; (all from Sigma-Aldrich). The mixture was pre-incubated for 10 min at 30 °C under mild agitation. Afterwards 50 µL of potato starch solution (1% stock solution in water, *w*/*v*) was added and incubated for 10 min. Finally, the reaction was terminated by adding 250 μL of color reagent mixture (96 mM 3,5-dinitrosalicylic acid solution prepared in 5.3 M sodium potassium tartrate solution and 2 M NaOH,) and heating for 10 min at 99 °C. After cooling to room temperature, the mixture was diluted with 0.5 ml of ddH_2_O, and the absorbance was measured at 540 nm. In all assays, enzyme activity without inhibitor (negative control) and complete inhibition of enzymes (positive control) were measured by replacing peptides with water or acarbose (for α-amylase), phenylmethylsulfonylfluorid (for α-chymotrypsin) and sunflower trypsin inhibitor I (for trypsin), respectively. For each sample, the background signal was measured by replacing the enzyme with water. All the assays were carried out in three biological replicates and are presented as percentage of inhibition (mean ± standard deviation) relative to the total enzyme activity. Molar concentration of citcol peptides was determined by measuring absorption at 280 nm using Lamber-Beer’s law and the molar extinction coefficients for tryptophan (5690 M^−1^ cm^−1^), cystines (120 M^−1^ cm^−1^) and tyrosine (1280 M^−1^ cm^−1^).

### 2.7. Sequence Homology Analysis

For initial online homology blastp and tblastn searches against UniProt/NCBI (National Center for Biotechnology Information) databases, the amino acid sequences of citcol peptides were used as queries and results with E-values of less than 0.1 were further considered. In addition, representative annotated peptide sequences of different functional CRP families, comprising at least six Cys residues in the mature processed peptide domain and belonging to different CXC patterns were used for comparison to novel citcol peptides: *Amaranthus hypochondriacus* (UniProt entry: P80403), *Spinacia oleracea* (P84781), *Phytolacca americana* (P81418), *Solanum tuberosum* (P01075), *Astragalus sinicus* (Q07A30), *Amaranthus retroflexus* (Q5I2B2), *Momordica cochinchinensis* (P82410). For similarity analysis of the initial hits a pairwise sequence comparison was carried out using the EMBOSS Needle online tool with citcol-8 as reference. For in-depth blastp searches, plant genomes and transcriptomes were downloaded from NCBI’s FTP server, and additional 1341 transcriptomes (Data S1) were retrieved from the 1-kp project [33]. Prior to blastp analysis all data was six-frame translated using Python and Biopython [34]. As queries we used the following protein sequences: citcol 1-8, P01074, P10291, P10292, P10293, P10295, P11969, Q9S8D2 and Q9S8W3. Hits with E-values of less than 0.001 were considered as significant. Multiple sequence alignments were performed by Multiple Sequence Comparison by Log-Expectation (MUSCLE) [35], and sequence logos were generated using WebLogo application [36]. The phylogenetic tree construction and pairwise similarity analysis were performed by maximum-likelihood method with sequences found by blastp searches against six-frame translated genomes/transcriptomes with the software MEGA X [37], version 10.1.7, respectively and illustrated by iTOL [37], version 5.5.

## 3. Results and Discussion

Nature-derived peptides often exhibit high stability based on a network of disulfide bonds while maintaining great tolerance for sequence modifications of the intercysteine loops, a feature termed ’structural plasticity’ [38]. Consequently, CRPs are under investigation as molecular scaffolds for engineering pharmaceutically useful peptides. CRPs have been studied from numerous plant families [39,40], and the squash family (Cucurbitaceae) is known as a rich source of these stabilized peptides [5,41]. In this study, we isolated and characterized novel CRPs from *C. colocynthis* a member of the Cucurbitaceae family. We confirmed the presence of at least 23 CRPs in this plant based on peptide mass fingerprinting, which were de novo sequenced by enzymatic digests, mass spectrometry and amino acid analysis. Based on sequence homology analysis we predicted the putative cysteine-connectivity of the novel citcol peptides. Genome/transcriptome mining and a subsequent bioinformatics analysis enabled us to elucidate the phylogenetic distribution and to propose a classification of Cucurbitaceae CRPs into trypsin inhibitors and antimicrobial peptides providing a better understanding of the distribution of these two subgroups of the manifold ‘knottins’ peptide family.

### 3.1. Preparation of Peptide-Enriched Citcol Extracts

Since Cucurbitaceae plants are known as a rich source for CRPs we explored the proteome of C. colocynthis for discovering novel CRPs. To remove secondary metabolites and plant pigments such as chlorophyll, 400 g of dried plant material were extracted overnight by dichloromethane/MeOH. After filtration, addition of water and thorough liquid/liquid phase partitioning, the aqueous phase containing peptides, which yielded 430 mg crude dried material was used for subsequent experiments. Removal of polar plant compounds was achieved by C_18_ solid-phase extraction in batch, while washing and eluting with increasing concentrations of acetonitrile (solvent B). Molecular weight analysis of the collected fractions obtained one sample with an elution condition of 35%–70% solvent B as major source of the desired peptides within a mass range of 3.5–4.5 kDa. Chemical modification of cysteine residues via alkylation confirmed the presence of at least 23 unique peptide mass signals with a mass shift of 348 Da, each corresponding to six carbamidomethylated cysteine residue (Figure 1), suggesting the citcol peptides may belong to a CRP family comprising three disulfide bonds.

### 3.2. Reversed-Phase HPLC Fractionation and Purification

RP-HPLC was performed for purification of detected CRPs from the peptide-enriched extract of *C. colocynthis*. MALDI-TOF analysis of fractions collected from RP-HPLC enabled the identification of peptide peaks in the chromatogram, which typically eluted between 42 and 55% solvent C (Figure 2A). Preparative and semi-preparative RP-HPLC allowed isolation of eight different CRPs with mass signals between *m/z* 3658.1 to 4160.3, which were named citcol-1 to -8 (*Citrullus colocynthis*) (Figure 2B). Analytical HPLC and MALDI-TOF analysis were used to confirm purity of the isolated citcol peptides ≥90% for individual peptides (exemplarily shown for citcol-2 in Figure 2C,D).

### 3.3. De Novo Sequencing of Citcol Peptides by MALDI-TOF/TOF

To determine the amino acid sequences of citcol peptides, well-established peptidomics protocols were applied [14]. Based on MALDI-TOF analysis, S-carbamidomethylation, resulted in a mass shift of +347.9±0.2 Da corresponding to the presence of six cysteine residues in citcol peptides (Figure 1). To determine complete sequence of each peptide the Cys-modified peptides were partially digested by trypsin and chymotrypsin to produce short peptide fragments which are generally well amenable to collision-induced fragmentation. Exemplarily, the predicted cleavage sites and mass spectra obtained after digestion of citcol-2 are shown in Figure 3A–C. There were three major mass signals with an absolute intensity >500 after site specific chymotryptic proteolysis with corresponding monoisotopic masses of 1246.6 Da, 1723.8 Da, 1858.6 Da, and three major fragments with 822.4 Da, 1435.7 Da, and 3279.2 Da resulted by tryptic digestion. The identified peaks of citcol-2 were used as precursor ions for collision-induced fragmentation and sequence assembly by assignment of N-terminal b- and C-terminal y-ion series (Figure 3D–G, Appendix A). Using sequence fragment assembly, we identified the correct order of overlapping fragments and full-length sequences of citcol peptides. For instance, citcol-2 has the sequence NH_2_-VCLFVGKPCWSDADCPSGCYCKPLPLIDAGYCGFL-COOH. Complete tryptic/chymotryptic digestion allowed to distinguish between isobaric residues Gln/Lys and Leu/Ile, respectively and sequence similarity analysis, high-sensitivity amino acid analysis and automated ion fragmentation calculation confirmed the results obtained by de novo peptide sequencing. The sequences of other citcol peptides were elucidated in the same way and the mass spectrometry data (Appendix A) and sequences of these peptides are summarized in Table 1.

Our peptidomics approach does not allow a definite conclusion as to whether all citcol peptides are translated from individual genes, or whether they are produced by splicing events or via posttranslational processing, but there are examples from other plants showing that indeed truncated analogs of CRPs can be made by alternative splicing [42], which has also been reported for other proteins [43,44,45,46,47]. Alignment of citcol-1 to -8 showed that they have a conserved cysteine spacing with the structure X_1-5_CX_6_CX_5_CX_3_CX_1_CX_10_CX_3_ in which X represents any amino acid residue except cysteine (Figure 4A). By sequence similarity to other characterized Momordica peptides, we predict the citcol peptides to have the following putative cysteine connectivity C_I_-C_IV_, C_II_-C_V_, C_III_-C_VI_ (Appendix A), which is yet to be confirmed in a future study.

### 3.4. Genome Mining and Sequence Homology Analysis

To obtain information on the putative biological function of citcol peptides, we developed custom bioinformatics approaches. Standard database searching and sequence homology analysis against UniProt (via blastp) using citcol peptide sequences as queries did not yield any hits with significant homology (E-value ≤ 0.1) to functional characterized peptides. For instance, in our search against UniProt, five hits, which are classified as uncharacterized proteins, were found (A0A0A0KY29, A0A5A7TRX6, A0A5D3E1P7, A0A5A7TM06, A0A5D3E1J3). Additional mining of deposited genome/transcriptome data using the tblastn algorithm yielded a hit with identity to citcol-8 in the translated chromosome 10 of *Citrullus lanatus* (GenBank: VOOL01000010.1) and one hit differing in six amino acids in the translated chromosome 7 of the genome of *Cucumis melo* (GenBank: LN713261.1) (Table 2).

UniProt/NCBI database analysis by blastp search was used to understand the similarity of citcol peptides to other functional peptide families and annotated CRPs. Initially this search provided one functionally annotated hit, i.e., a trypsin inhibitor peptide isolated from *Momordica cochinchinensis* (E-value >0.1). To evaluate the similarities and Cys loop pattern of citcol peptides with other functional CRPs (or precursors thereof), we selected representative cysteine-rich peptides and sequences sporadically from different families based on functionality and comprising at least six Cys residues (Table 3). A pairwise distance analysis was performed against the following sequences: an α-amylase inhibitor (from *Amaranthus hypochondriacus*), two trypsin inhibitors (from *Momordica cochinchinensis* and *Spinacia oleracea*), an anti-fungal/antimicrobial peptide (from *Amaranthus retroflexus*), a metallocarboxypeptidase inhibitor (from *Solanum tuberosum*), a nodule-specific protein (from *Astragalus sinicus*), and an antimicrobial peptide (from *Phytolacca americana*) (Table 3). Homology analysis revealed that the trypsin inhibitor peptide MCoTI-III isolated from *Momordica cochinchinensis* (Cucurbitaceae) shares the highest, i.e., 43.6% sequence homology with citcol-8. In addition, sequence comparison of citcol peptides with several other trypsin inhibitors from *Momordica spp.* (namely MCoTI-I, -II, -III and EETI-II, Appendix A) [48,49,50] revealed that loops 1 to 4 of these peptides share a similar length. Therefore, to determine the functional properties of novel citcol peptides, we tested their protease inhibitory activity. We assayed different concentrations of pure citcol peptides for inhibition of bovine trypsin and α-chymotrypsin. Furthermore, we also tested the inhibitory activity against α-amylase enzymes of *Aspergillus oryzae, Bacillus subtilis* and *Sus sucrofa*. Surprisingly, the citcol peptides did not exhibit any inhibitory activity against these enzymes (Appendix A), despite their apparent similarity to functional enzyme inhibitor peptides (Table 3). To understand this functional discrepancy, i.e., why citcol peptides do not inhibit enzyme activity (based on the tested assays), a detailed genome/transcriptome mining approach was performed using sequence homology analysis to other squash family (Cucurbitaceae) CRPs. This detailed analysis yielded 97 sequences (Data 2) that were homologous to citcol peptides (E-value ≤ 0.1) from several sub-families (genus) of Cucurbitaceae, including Cucurbita, Cucumis, Luffa, Lagenaria and Momordica. The phylogenetic distribution of these publicly available sequences revealed that they cluster into two distinct branches: an antimicrobial and a trypsin inhibitory branch (Figure 4A). Furthermore, the discovered citcol peptides are more similar to antimicrobial peptides than trypsin inhibitory peptides (Figure 4B), as indicated for instance by the length of loop 5.

At the molecular detail, we noticed a minor but apparently important modification in the sequence of citcol peptides: comparison of squash family trypsin inhibitor peptides isolated from the seeds of Cucurbitaceae showed that the two basic amino acids Arg and Lys, respectively, are important residues responsible for trypsin inhibitory activity in the active site (P_1_ position) in loop 1 (Appendix A), whereas in citcol peptides the P_1_ position has been occupied by a Phe residue. This could be one reason for the observed lack of trypsin inhibitory activity of citcol peptides. Nevertheless, such a mutation may not explain the lack of activity against chymotrypsin and α-amylase. For instance in Bowman-Birk inhibitors a Lys to Phe mutation reportedly was sufficient for a functional change from trypsin to chymotryptic inhibitory capacity [57]. Therefore, it is likely that other modifications, such as the observed differences in loop 5 of citcol peptides as compared to CRPs of *Momordica spp.* (Figure 4A, Appendix A) are contributing factors to the observed lack of enzyme inhibition.

## 4. Conclusions

There are several reports attributing diverse biological activities to plant-derived CRPs [58,59,60], and in silico analysis such as genome mining has had an important role in CRP discovery [61]. In fact, CRPs have also been found in other organisms, including humans [62] and animals [63,64]. Isolation and characterization of peptides with novel sequences and/or cysteine motifs or new functions is an important process in the discovery of novel potential pharmaceutics. Here, we purified and characterized eight novel CRPs (citol-1 to citcol-8) from *C. colocynthis*. Several plant peptides that share sequence similarity to the here-described citcol peptides are well-known protease inhibitors [5,17,48,49]. However, detailed homology and phylogenetic analysis of these citcol peptides in comparison to other Cucurbitaceae CRPs suggested that they were a novel subtype of CRPs with minor variations in loops 1 and 5, leading to a functional specialization, since these novel citcol CRPs did not inhibit representative protein or sugar processing enzymes. For instance, it has been shown that a single amino acid substitution in loop 1 of inhibitory peptides can alter their activity and specificity or inactivate their function [65,66]. It is likely that such minor molecular changes led to the loss of enzyme inhibitory activity, as confirmed in several inhibition assays. At a more general level, the apparent sequence similarities between different subfamilies of plant “trypsin inhibitor” peptides may be misleading for a correct functional classification. Our results underline that detailed bioinformatics analysis is an expedient strategy applicable for discovery of cysteine-rich plant peptides. Diversity in the intercysteine loop sequences despite a common cysteine pattern suggest additional bioactivities for these CRPs that are yet to be identified. Overall, our study adds to the biodiversity of CRPs, a family of plant biomolecules that experiences increasing attention for applications in drug design and agriculture [6,67].

## Figures and Tables

**Figure 1 biomolecules-10-01326-f001:**
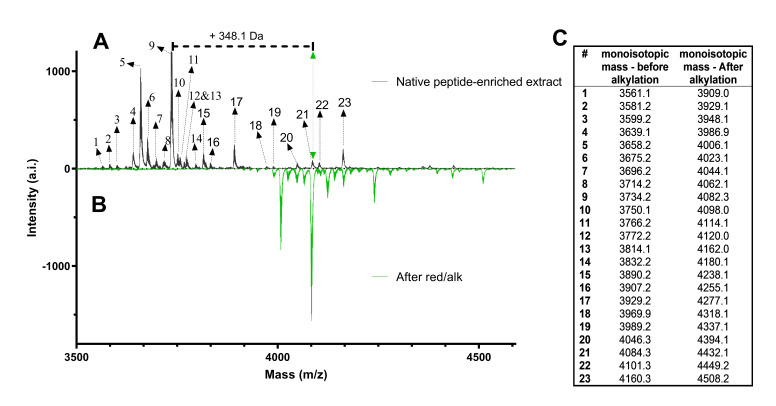
Chemical analysis of the citcol peptidome using matrix-assisted laser desorption ionization time-of-flight (MALDI-TOF) analysis. (**A**) The mass spectrum of a representative *Citrullus colocynthis* C_18_ peptide-enriched extract is illustrated in the range of *m/z* 3500–4600 (black trace). Each black dotted arrow indicated by numbers (#) represents a cysteine-rich peptide (refer to Table in C). (**B**) Chemical modification of cysteine residues was performed by reduction and alkylation. A representative spectrum after S-carbamidomethylation is illustrated in green. The mass shift of Δ348 Da indicates the presence of six cysteine residues. For instance, the peptide signal at *m/z* 3734.2 shifts by 348.1 Da to *m/z* 4082.3. (**C**) List of monoisotopic mass signals [M + H]^+^ of citcol peptides before and after S-carbamidomethylation.

**Figure 2 biomolecules-10-01326-f002:**
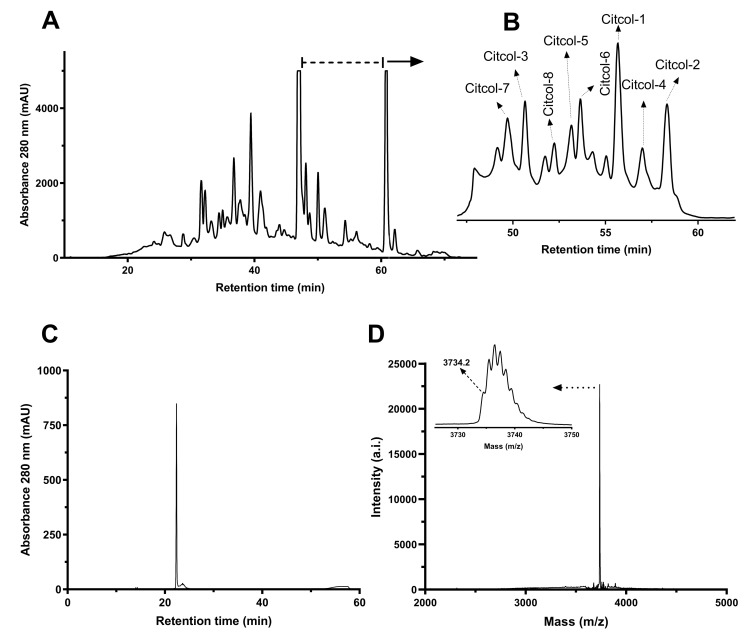
Fractionation and purification of cysteine-rich citcol peptides by reversed-phase HPLC. (**A**) Purification chromatogram by preparative RP-HPLC; the dashed line indicates the elution region of citcol peptides. (**B**) Semi-preparative RP-HPLC of the elution region of citcol peptides (retention time 47-60 min) indicating peaks for citcol-1 to -8. (**C**) Analytical RP-HPLC chromatogram of citcol-2 (purity ~95%). (**D**) Molecular weight analysis of citcol-2 corresponding to a monoisotopic mass [M + H]^+^ of 3734.1 Da. For HPLC analysis linear gradients of solvent C (90% acetonitrile, 9.92% H_2_O and 0.08% TFA) were used at flow rates of 8, 3 and 0.3 mL/min for preparative, semi-preparative and analytical RP-HPLC, respectively.

**Figure 3 biomolecules-10-01326-f003:**
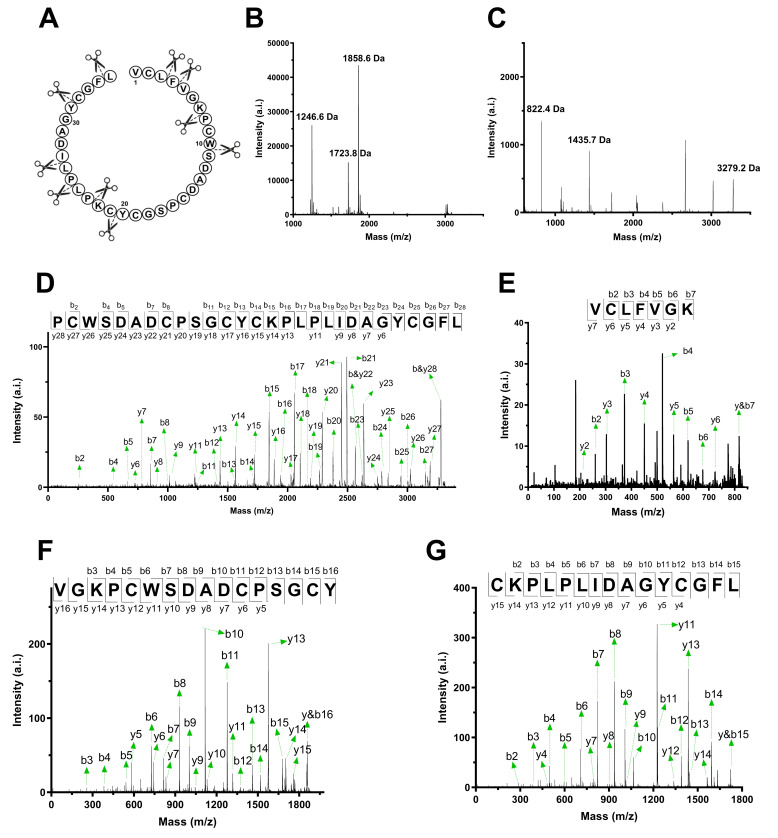
De novo sequencing of citcol-2 by MALDI-TOF/TOF. (**A**) Schematic structure and full sequence of citcol-2; scissors on the inside of the circle indicates potential trypsin cleavage sites and scissors on the outside indicate potential chymotrypsin cleavage sites. MS spectrum of citcol-2 after partial digestion with chymotrypsin (**B**) and trypsin (**C**). MS/MS fragmentation spectra of precursor peptides with a molecular weight of 3279.2 Da (**D**) and 822.4 Da (**E**) after tryptic digestion, and 1858.6 Da (**F**) and 1723.8 Da (**G**) after chymotrypsin digestion, which covers the entire sequence of citcol-2. The amino acid sequence was determined by assembly of digested fragments and assignment of N-terminal b- and C-terminal y-ions series. Complete tryptic digestion was used to distinguish between isobaric amino acids Gln/Lys and complete chymotrypsin for Leu/Ile. Comparison of the sequences of the fragments produced by these two enzymes showed the correct order of the fragments. The *m/z* value list of each of the precursor masses used for de novo sequencing are shown in Appendix A, where A, B, C and D correspond to precursors with a molecular weight of 3279.2 Da, 822.4 Da, 1723.8 Da and 1858.6 Da, respectively.

**Figure 4 biomolecules-10-01326-f004:**
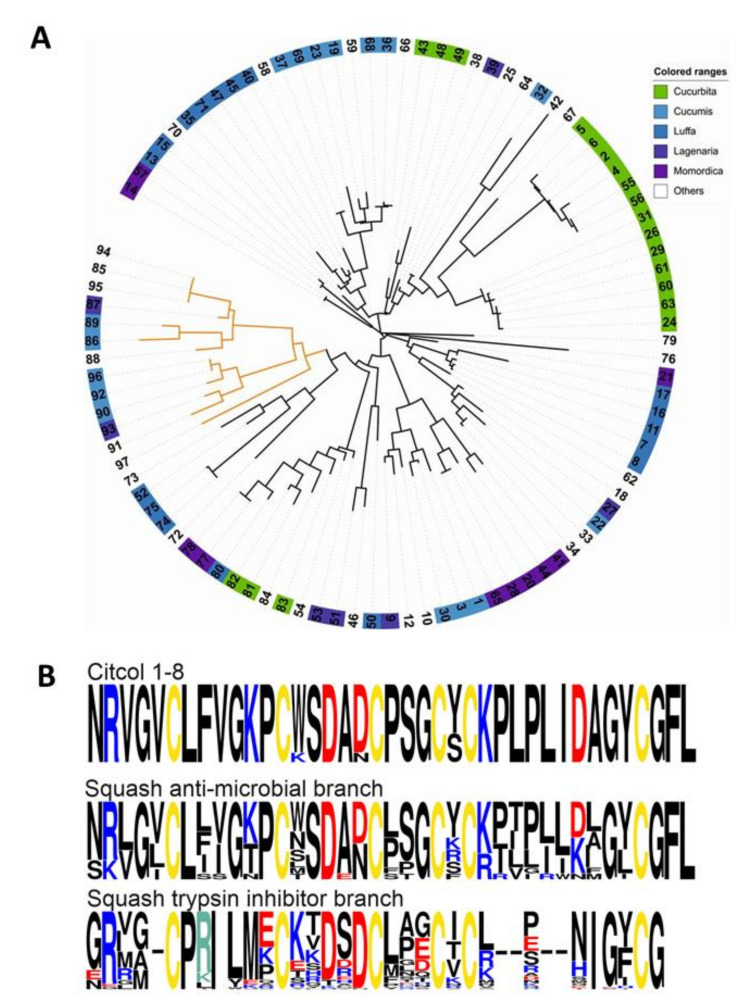
Homology and phylogenetic distribution of citcol peptides. (**A**) Maximum-likelihood phylogenetic tree calculated with sequences found by blastp searches against six-frame translated genomes/transcriptomes (E-value ≤ 0.1). The antimicrobial branch is colored in orange and the trypsin inhibitory branch is colored in black. The sequences are encoded by numbers (Data 2), and the Cucurbitaceae subfamilies summarized with one color (Cucurbita—green, Cucumis—light blue, Luffa—dark blue, Lagenaria—dark purple, Momordica—violet, Others—white). (**B**) Alignment of frequency logos calculated with citcol-1 to -8, sequences from the squash antimicrobial peptides branch and the squash trypsin inhibitor branch (see Materials and Methods for details). Cysteines are colored in yellow, amino acids with positively charged side chains (at pH 7) are colored in blue (H, K, R) and with negative charged sidechains (at pH 7) are colored in red (D, E). Gaps ‘-’ were introduced to maximize the alignment.

**Table 1 biomolecules-10-01326-t001:** Properties and sequence comparison of isolated citcol peptides.

Name	[M + H]^+1^	Sequence ^2^	Length
		loop 1 2 3 4 5	
citcol-1	3658.1	----VCLFVGKPCWSDADCPSGCSCKPLPLIDAGYCGFL	35 aa
citcol-2	3734.1	----VCLFVGKPCWSDADCPSGCYCKPLPLIDAGYCGFL	35 aa
citcol-3	3814.4	--VGVCLFVGKPCWSDADCPSGCSCKPLPLIDAGYCGFL	37 aa
citcol-4	3890.2	--VGVCLFVGKPCWSDADCPSGCYCKPLPLIDAGYCGFL	37 aa
citcol-5	4046.9	-RVGVCLFVGKPCWSDADCPSGCYCKPLPLIDAGYCGFL	38 aa
citcol-6	4084.1	NRVGVCLFVGKPCWSDADCPSGCSCKPLPLIDAGYCGFL	39 aa
citcol-7	4101.2	NRVGVCLFVGKPCKSDANCPSGCYCKPLPLIDAGYCGFL	39 aa
citcol-8	4160.3	NRVGVCLFVGKPCWSDADCPSGCYCKPLPLIDAGYCGFL	39 aa
		********* ***:***** ***************	

^1^ monoisotopic mass signal; ^2^ Conserved and similar residues in all sequences have been marked by asterisks and colons below the bottom line, respectively. All sequences are in N- to C-terminal direction. asterisk ’*’ and colon ’:’ indicate identical and similar residues in the alignment.

**Table 2 biomolecules-10-01326-t002:** Homologous sequences of citcol peptides found by a tblastn search against Genbank ^1^.

Species	Sequence ^2^	E-Value ^3^	Accession No
*C. colocynthis* (citcol-8)	NRVGVCLFVGKPCWSDADCPSGCYCKPLPLIDAGYCGFL	n.a.	this work
*Citrullus lanatus* (chr10 ^4^)	NRVGVCLFVGKPCWSDADCPSGCYCKPLPLIDAGYCGFL	3 × 10^−23^	VOOL01000010.1
*Cucumis melo* (chr7 ^4^)	-RLGVCLLVGKPCMSDADCPSGCYCKPVPLLDIGYCGFL	5 × 10^−16^	LN713261.1
	*:****:***** *************:**:* ******		

^1^ only sequences containing six cysteine residues are reported; ^2^ conserved and similar residues in all sequences have been marked by asterisks and colons below the bottom line, respectively. ^3^ E-value cut-off of 0.001 was applied; ^4^ chr: chromosome. n.a.—not applicable, asterisk ‘*’ and colon ‘:’ indicate identical and similar residues in the alignment.

**Table 3 biomolecules-10-01326-t003:** Comparison of citcol-8 with representative plant cysteine-rich sequences and peptides.

Species	Sequence	Uniprot ID	Function	Cys Loops Pattern	Similarity (%) ^1^	Reference
*C. colocynthis* (citcol-8)	NRVGVCLFVGKPCWSDADCPSGCYCKPLPLIDAGYCGFL	n.a.	n.a.	CX_6_CX_5_CX_3_CX_1_CX_10_C ^2^	n.a.	this work
*Momordica cochinchinensis*	QRACPRILKKCRRDSDCPGECICKENGYCG	P82410	trypsin inhibitor	CX_6_CX_5_CX_3_CX_1_CX_5_C	43.6	[51]
*Solanum tuberosum*	HADPICNKPCKTHDDCSGAWFCQACWNSARTCGPY	P01075	metallocarboxy peptidase inhibitor	CX_3_CX_5_CX_5_CX_2_CX_6_C	38.5	[52]
*Phytolacca americana*	AGCIKNGGRCNASAGPPYCCSSYCFQIAGQSYGVCKNR	P81418	antimicrobial peptide	CX_6_CX_8_CCX_3_CX_10_C	31.7	[53]
*Spinacia oleracea*	EDKCSPSGAICSGFGPPEQCCSGACVPHPILRIFVCQ	P84781	trypsin inhibitor	CX_6_CX_8_CCX_3_CX_10_C	29.5	[54]
*Amaranthus hypochondriacus*	CIPKWNRCGPKMDGVPCCEPYTCTSDYYGNCS	P80403	α-amylase inhibitor	CX_6_CX_8_CCX_4_CX_7_C	27.3	[55]
*Amaranthus retroflexus*	AGECVQGRCPSGMCCSQFGYCGRGPKYCGR	Q5I2B2	antimicrobial peptide	CX_4_CX_4_CCX_5_CX_6_C	26.7	[17]
*Astragalus sinicus*	TYSCGGHIDCKDFCKSEGYRGFKCTPKKTCTCFH	Q07A30	nodule specific protein	CX_5_CX_3_CX_9_CX_5_CX_1_C	20.0	[56]

^1^ the percentage of similarity was calculated based on pairwise alignment of citcol-8 as reference using the EMBOSS Needle online tool; ^2^ X represents any amino acid residue except cysteine. n.a.—not applicable.

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
