# Peer review of "Isolation of Cysteine-Rich Peptides from *Citrullus colocynthis"

_biomolecules, 2020, doi:10.3390/biom10091326_

Round 1

Reviewer 1 Report

Major comments:

line 193: why did the authors choose these very specific sequences for the phylogenetic tree? This should need a more clear explanation as they have only chosen one per plant species, and for example the Arabidopsis peptide sequence, although rich in cysteine residues (it has 24), has 806 aminoacids so it is not a small peptide. This is not straightforward to understand. 

Minor comments:

Abstract: is missing the concept of the relevance of small molecules in the pharmaceutic market.

line 87: peptides were enriched

line 193: why did the authors choose these very specific sequences for the phylogenetic tree?

line 295: Use capital X for the consensus peptide sequence

line 299: autors should say putative biological function as the approach is using bioinformatics

line 312: functional  families. And also: the selected Arabidopsis peptide has more than 6 cysteines.

Author Response

Reviewer 1:

Comments and Suggestions for Authors

Major comments:

line 193: why did the authors choose these very specific sequences for the phylogenetic tree? This should need a more clear explanation as they have only chosen one per plant species, and for example the Arabidopsis peptide sequence, although rich in cysteine residues (it has 24), has 806 amino acids so it is not a small peptide. This is not straightforward to understand.

Response: We appreciate this constrictive comment. We have revised and added the following explanation for clarity:

  • For initial online homology blastp and tblastn searches against UniProt/NCBI (National Center for Biotechnology Information) databases, the amino acid sequences of citcol peptides were used as queries and results with E-values of less than 0.1 were further considered. In addition, representative annotated peptide sequences of different functional CRP families, comprising at least six Cys residues in the mature processed peptide domain and belonging to different CXC patterns were used for comparison to novel citcol peptides: Amaranthus hypochondriacus (UniProt entry: P80403), Spinacia oleracea (P84781), Phytolacca americana (P81418), Solanum tuberosum (P01075), Astragalus sinicus (Q07A30), Amaranthus retroflexus (Q5I2B2), Momordica cochinchinensis (P82410). For similarity analysis of the initial hits a pairwise sequence comparion was carried using the EMBOSS Needle online tool with citcol-8 as reference. For in-depth blastp searches, plant genomes and transcriptomes were downloaded from NCBI´s FTP server, and additional 1341 transcriptomes (Supplementary Data 1) were retrieved from the 1-kp project [33]. Prior to blastp analysis all data was six-frame translated using Python and Biopython [34]. Hits with E-values of less than 0.001 were considered as significant. Multiple sequence alignments were performed by Multiple Sequence Comparison by Log-Expectation (MUSCLE) [35], and sequence logos were generated using WebLogo application [36]. The phylogenetic tree construction and pairwise similarity analysis were performed by maximum-likelihood method with sequences found by blastp searches against six-frame translated genomes/transcriptomes with the software MEGA X [37], version 10.1.7,respectively and illustrated by iTOL [37], version 5.5.” (lines 189-207) and
  • UniProt/NCBI database analysis by blastp search was used to understand the similarity of citcol peptides to other functional peptide families and annotated CRPs. Initially this search provided one functionally annotated hit, i.e. a trypsin inhibitor peptide isolated from Momordica cochinchinensis (E-value >0.1). To evaluate the similarities and Cys loop pattern of citcol peptides with other functional CRPs (or precursors thereof), we selected representative cysteine-rich peptides and sequences sporadically from different families based on functionality and comprising at least six Cys residues (Table 3).” (lines 328-334)

In addition, we have deleted the Arabidopsis sequence from Table 3, since it is yet to be confirmed whether the large precursor is in fact being processed into small CRPs.

Minor comments:

Abstract: is missing the concept of the relevance of small molecules in the pharmaceutic market.

Response: We added the following information to the abstract: “Based on the ancient knowledge about the healing properties of herbal preparations, plant-derived small molecules, e.g. salicylic acid, or quinine, have been integral to modern drug discovery. Besides, many plant families, such as Cucurbitaceae, are known as a rich source for cysteine-rich peptides, which are gaining importance as valuable pharmaceuticals.” (lines 13-16)

line 87: peptides were enriched

Response: Thank you for pointing out this spelling mistake; it has been corrected.

line 193: why did the authors choose these very specific sequences for the phylogenetic tree?

Response: As pointed out above (and the manuscript), we performed sequence homology analysis using publicly available genome and transcriptome datasets, focusing on the squash family: “To understand this functional discrepancy i.e. why citcol peptides do not inhibit enzyme activity (based on the tested assays), a detailed genome/transcriptome mining approach was performed using sequence homology analysis to other squash family (Cucurbitaceae) CRPs. This detailed analysis yielded 97 sequences (Supplementary Data 2) that are homologous to citcol peptides (E-value ≤ 0.1) from several sub-families (genus) of Cucurbitaceae, including Cucurbita, Cucumis, Luffa, Lagenaria and Momordica. The phylogenetic distribution of these publicly available sequences revealed that they cluster into two distinct branches: an antimicrobial and a trypsin inhibitory branch (Figure 4A). Furthermore, the discovered citcol peptides are more similar to antimicrobial peptides than trypsin inhibitory peptides (Figure 4B), for instance indicated by the length of loop 5.” (lines 349-373)

line 295: Use capital X for the consensus peptide sequence

Response: As suggested we have now used capital X for all consensus sequences, in the text and Tables.

line 299: autors should say putative biological function as the approach is using bioinformatics

Response: Biological function has been clarified to as “putative”. (line 316)

line 312: functional  families. And also: the selected Arabidopsis peptide has more than 6 cysteines.

Response: To avoid confusion we have deleted the word “functional”, since it was obsolete in this context (line 575). Furthermore, the Arabidopsis sequence has been excluded from the similarity analysis (Table 3).

Reviewer 2 Report

The authors aimed to identify and characterize cysteine-rich peptides from Citrullus colocynthis, using techniques such as small peptides purification, sequencing and genome data mining. The novelty of the paper is the identification of 23 CRPs in this species. Some of them lack the protease inhibitor activity, although their 48 % sequence similarity with Momordica spp trypsin inhibitor. The manuscript is in the scope of Biomolecules, but revisions are needed. 

1) The authors should improve their discussion, confronting their results with the vast literature in the area of CRP. 

2) The weak part of the paper is the genome/transcriptome data mining. It was not clear how the authors used this data to retrieve CRP sequence information.

3) The authors need to clarify how they constructed the phylogenetic tree, neighbor-joining or maximum likelihood method. There is a conflict in the manuscript. Besides, no explanation about the phylogenetic analysis was found in the text.

3)The authors need to pay attention on figure citations on the text (e.g. Figure 4B was not cited).

Author Response

Reviewer 2:

The authors aimed to identify and characterize cysteine-rich peptides from Citrullus colocynthis, using techniques such as small peptides purification, sequencing and genome data mining. The novelty of the paper is the identification of 23 CRPs in this species. Some of them lack the protease inhibitor activity, although their 48 % sequence similarity with Momordica spp trypsin inhibitor. The manuscript is in the scope of Biomolecules, but revisions are needed.

Response: We thank this reviewer for her/his constructive comments, that we have addressed (see below) to improve the manuscript.

1) The authors should improve their discussion, confronting their results with the vast literature in the area of CRP.

Response: We agree and have now revised the discussion at several levels; we added relevant information regarding other CRPs, and discussed our study with some representative manuscripts from the vast amount of CRP literature, for instance:

CRPs have been studied from numerous plant families [39,40], and the squash family (Cucurbitaceae) is known as a rich source of these stabilized peptides [5,41].” (line 212-214)

There are several reports of attributing diverse biological activities to plant derived CRPs [58-60] and in silico analysis such as genome mining have had an important role in CRP discovery [61]. In fact, CRPs have been also found in other organisms including humans [62] and animals [63,64]. Isolation and characterization of peptides with novel sequences and/or cysteine motifs, or new functions is an important process in the discovery of novel potential pharmaceutics. Here, we purified and characterized eight novel CRPs (citol-1 to citcol-8) from C. colocynthis.” (lines 588-593)

Added References: 39-41 and 58-64

2) The weak part of the paper is the genome/transcriptome data mining. It was not clear how the authors used this data to retrieve CRP sequence information.

Response: For clarity we have revised the Methods section and added further information to the Results:

  • For initial online homology blastp and tblastn searches against UniProt/NCBI (National Center for Biotechnology Information) databases, the amino acid sequences of citcol peptides were used as queries and results with E-values of less than 0.1 were further considered. In addition, representative annotated peptide sequences of different functional CRP families, comprising at least six Cys residues in the mature processed peptide domain and belonging to different CXC patterns were used for comparison to novel citcol peptides: Amaranthus hypochondriacus (UniProt entry: P80403), Spinacia oleracea (P84781), Phytolacca americana (P81418), Solanum tuberosum (P01075), Astragalus sinicus (Q07A30), Amaranthus retroflexus (Q5I2B2), Momordica cochinchinensis (P82410). For similarity analysis of the initial hits a pairwise sequence comparion was carried using the EMBOSS Needle online tool with citcol-8 as reference. For in-depth blastp searches, plant genomes and transcriptomes were downloaded from NCBI´s FTP server, and additional 1341 transcriptomes (Supplementary Data 1) were retrieved from the 1-kp project [33]. Prior to blastp analysis all data was six-frame translated using Python and Biopython [34]. Hits with E-values of less than 0.001 were considered as significant. Multiple sequence alignments were performed by Multiple Sequence Comparison by Log-Expectation (MUSCLE) [35], and sequence logos were generated using WebLogo application [36]. The phylogenetic tree construction and pairwise similarity analysis were performed by maximum-likelihood method with sequences found by blastp searches against six-frame translated genomes/transcriptomes with the software MEGA X [37], version 10.1.7,respectively and illustrated by iTOL [37], version 5.5.” (lines 189-207) and
  • UniProt/NCBI database analysis by blastp search was used to understand the similarity of citcol peptides to other functional peptide families and annotated CRPs. Initially this search provided one functionally annotated hit, i.e. a trypsin inhibitor peptide isolated from Momordica cochinchinensis (E-value >0.1). To evaluate the similarities and Cys loop pattern of citcol peptides with other functional CRPs (or precursors thereof), we selected representative cysteine-rich peptides and sequences sporadically from different families based on functionality and comprising at least six Cys residues (Table 3).” (lines 328-334)

3) The authors need to clarify how they constructed the phylogenetic tree, neighbor-joining or maximum likelihood method. There is a conflict in the manuscript. Besides, no explanation about the phylogenetic analysis was found in the text.

Response: We used the maximum likelihood method to construct the tree. We apologize for this misunderstanding, and have now been clarified the methodology and results, accordingly (see point 2 above). We have also added explanation of the sequence homology analysis: “To understand this functional discrepancy i.e. why citcol peptides do not inhibit enzyme activity (based on the tested assays), a detailed genome/transcriptome mining approach was performed using sequence homology analysis to other squash family (Cucurbitaceae) CRPs. This detailed analysis yielded 97 sequences (Supplementary Data 2) that are homologous to citcol peptides (E-value ≤ 0.1) from several sub-families (genus) of Cucurbitaceae, including Cucurbita, Cucumis, Luffa, Lagenaria and Momordica. The phylogenetic distribution of these publicly available sequences revealed that they cluster into two distinct branches: an antimicrobial and a trypsin inhibitory branch (Figure 4A). Furthermore, the discovered citcol peptides are more similar to antimicrobial peptides than trypsin inhibitory peptides (Figure 4B), for instance indicated by the length of loop 5.” (lines 349-373), and revised Figure caption 4: “Figure 4. Homology and phylogenetic distribution of citcol peptides. (A) Maximum-likelihood phylogenetic tree calculated with sequences found by blastp searches against six-frame translated genomes/transcriptomes (E-value ≤ 0.1). The antimicrobial branch is coloured in orange and the trypsin inhibitory branch is coloured in black. The sequences are encoded by numbers (Supplementary Data 2), and the Cucurbitaceae subfamilies summarized with one colour (Cucurbita - green, Cucumis – light blue, Luffa – dark blue, Lagenaria – dark purple, Momordica – violet, Others – white). (A) Alignment of frequency logos calculated with citcol-1 to -8, sequences from the squash antimicrobial peptides branch and the squash trypsin inhibitor branch (see Materials and Methods for details). Cysteines are coloured in yellow, amino acids with positively charged side chains (at pH 7) are coloured in blue (H, K, R) and with negative charged sidechains (at pH 7) are coloured in red (D, E). Gaps were introduced to maximize the alignment. (lines 375-385)

4)The authors need to pay attention on figure citations on the text (e.g. Figure 4B was not cited).

Response: We have now referred to Fig. 4B (line 373) in the text and Supplementary Data 2 (line 368). in the text.